

# Improving the gnomonic approach with the *gnomonicM* R-package to estimate natural mortality throughout different life stages

Josymar Torrejón-Magallanes[1], Enrique Morales-Bojórquez[2] and Francisco Arreguín-Sánchez[1]

[1] Instituto Politécnico Nacional, Centro Interdisciplinario de Ciencias Marinas, La Paz, Baja California Sur, México
[2] Centro de Investigaciones Biológicas del Noroeste, La Paz, Baja California Sur, México

Corresponding author
Enrique Morales-Bojórquez,
embojorq@prodigy.net.mx

## ABSTRACT

Natural mortality ($M$) is defined as the rate of loss that occurs in a fish stock due to natural (non-fishing) causes and can be influenced by density-dependent or density-independent factors. Different methods have been used to estimate $M$, one of these is the gnomonic approach. This method estimates $M$ rates by dividing the life cycle of a species into subunits of time that increase as a constant proportion of the time elapsed from birth up to the initiation of each subdivision. In this study, an improved gnomonic approach is proposed to estimate natural mortality throughout different life stages in marine stocks using the *gnomonicM* package written in R software. This package was built to require data about (i) the number of gnomonic intervals, (ii) egg stage duration, (iii) longevity, and (iv) fecundity. With this information, it is possible to estimate the duration and natural mortality ($M_i$) of each gnomonic interval. The *gnomonicM* package uses a deterministic or stochastic approach, the latter of which assesses variability in $M$ by assuming that the mean lifetime fecundity ($MLF$) is the main source of uncertainty. Additionally, the *gnomonicM* package allows the incorporation of auxiliary information related to the observed temporal durations of specific gnomonic intervals, which is useful for calibrating estimates of $M$ vectors. The *gnomonicM* package, tested via deterministic and stochastic functions, was supported by the reproducibility and verification of the results obtained from different reports, thus guaranteeing its functionality, applicability, and performance in estimating $M$ for different ontogenetic developmental stages. Based on the biological information of Pacific chub mackerel (*Scomber japonicus*), we presented a new case study to provide a comprehensive guide to data collection to obtain results and explain the details of the application of the *gnomonicM* package and avoid its misuse. This package could provide an alternative approach for estimating $M$ and provide basic input data for ecological models, allowing the option of using estimates of variable natural mortality across different ages, mainly for life stages affected by fishing. The inputs for the *gnomonicM* packages are composed of numbers, vectors, or characters depending on whether the deterministic or stochastic approach is used, making the package quick, flexible, and easy to use; this allows users to focus on obtaining and interpreting results rather than the calculation process.

## INTRODUCTION

Mortality includes all factors that reduce the abundance of a population, and mortality has both natural and non-natural sources. Natural mortality ($M$) is defined as the rate of loss in a fish stock from natural (non-fishing) causes; natural mortality can be influenced by density-dependent (e.g., predation, cannibalism, disease, size) or density-independent (e.g., oceanographic variables, food availability) factors (*Pauly, 1980*; *Hampton, 2000*; *Beamish & Mahnken, 2001*). Natural mortality is one of the most influential input parameters in population dynamics and stock assessment models and is usually assumed to be a constant value for all ages and across time. Nevertheless, there is overwhelming evidence that $M$ is likely to vary substantially during the life history of a species, and individuals usually diminish in $M$ from early stages to adults in a population depending on age, length, or weight (*Caddy, 1991*; *Caddy, 1996*; *Lorenzen, 1996*; *Wang & Haywood, 1999*) or over interannual or greater time scales (*Zheng, Murphy & Kruse, 1995*; *Chu, Chien & Lee, 2008*; *Johnson et al., 2014*).

Different methods have been used to estimate $M$. The procedures most commonly reported in the literature include direct methods such as tag-recapture and telemetry (*Hearn, Pollock & Brooks, 1998*; *Pine et al., 2003*; *Pollock, Jiang & Hightower, 2004*); estimations inside stock assessment or ecosystem models (*Walters, Christensen & Pauly, 1997*; *Wang & Liu, 2006*; *Lee et al., 2011*; *Crone et al., 2019*); and finally metapopulation methods. These last methods are based on empirical equations that incorporate observable life-history parameters for several species (e.g., growth rate, maximum age, sexual maturity) and environmental variables (e.g., mean sea surface temperature) (*Alverson & Carney, 1975*; *Chen & Watanabe, 1989*; *Jensen, 1996*; *Kenchington, 2014*; *Pauly, 1980*). Among these procedures, metapopulation methods are the most commonly used methods for estimating $M$ since they demand little information; however, the estimates obtained based on these approaches can be biased and are always subject to great uncertainty (*Vetter, 1988*; *Schnute & Richards, 1995*; *Zheng, 2003*).

*Caddy (1991)* and *Caddy (1996)* proposed an approach to calculating an indicative vector of natural death rates at a given age that satisfies population replacement; in this approach, the initial death rate is high, falls off steeply in the early months of life and plateaus later on. The statistical proposal allows the estimation of $M$ from the egg stage to the adult stage for short-lived species following a decreasing trend, as is assumed to occur throughout the life history of species. Later, the "gnomonic model" was modified and extended to long-lived species, including the addition of criteria for adjusting the number of gnomonic intervals to the duration of real-life stages, improving the biological sense of the approach, and incorporating variability in fecundity, thus providing estimates of uncertainty in the outputs (*Martínez-Aguilar, Arreguín-Sánchez & Morales-Bojórquez, 2005*). These changes increased the versatility and utility of the "gnomonic model" for estimating $M$ in any
marine and freshwater spawning species (*Giménez-Hurtado, Arreguín-Sánchez & Lluch-Cota, 2009*; *Martínez-Aguilar et al., 2010*; *González-Peláez et al., 2015*; *Aranceta-Garza et al., 2016*; *Romero-Gallardo et al., 2018*).

According to *Caddy (1991)* and *Caddy (1996)*, the "gnomonic model" estimates *M* values with biological and ecological sense; the statistical procedure is simple and increases the realism in the changes of *M* that occur during the life cycle of a given species. In this study, the gnomonic approach was improved by modifying some equations and the estimation procedure. Additionally, a package called *gnomonicM* was written in the R language, providing a quick and user-friendly tool with which to estimate *M* throughout the different life stages of marine species. To demonstrate its functionality, the package was applied to the data of previous studies, and the results obtained herein were compared with the results reported in the original studies (*Caddy, 1996*; *Martínez-Aguilar, Arreguín-Sánchez & Morales-Bojórquez, 2005*) and with those of other published works that used the gnomonic approach. Finally, a detailed new case study focusing on Pacific chub mackerel (*Scomber japonicus*) was presented, providing a guide for the entire process, from the data compilation to the estimation of *M* values.

## MATERIALS & METHODS

The gnomonic model constitutes a distinctive approach for estimating natural mortality; its strength is associated with the use of simple biological variables that can be obtained from experimental or documented data. The main advantage of the gnomonic model is its estimation of natural mortality values for the entire life cycle of marine organisms; this is a feature of scientific utility because knowledge about mortality patterns during early life stages (e.g., egg and larvae) is limited and has high uncertainty. In many cases, the mortality of species in these stages can be estimated only under rearing conditions that are often expensive. Thus, a freely accessible code was developed for obtaining natural mortality estimates based on an improved version, with a new mathematical simplification that allows an increased performance of the gnomonic model in comparison to the original proposal published by *Caddy (1991)* and *Caddy (1996)*.

### Gnomonic interval model and new features

According to *Caddy (1996)* and *Martínez-Aguilar, Arreguín-Sánchez & Morales-Bojórquez (2005)*, the gnomonic method is supported by a negative exponential function in which the independent variable is $\Delta_i$, representing the temporal duration of the *i*thgnomonic interval; for $i = 1, 2, 3, \ldots, n$, the equation is expressed as follows:

$$N_i = \begin{cases} MLF * e^{-(M_i * \Delta_i)}; \ for \ i = 1 \\ N_{i-1} * e^{-(M_i * \Delta_i)}; \ for \ i > 1 \end{cases} \quad (1)$$

where $M_i$ is the average value of the natural mortality rate, which integrates the declining death rate through $\Delta_i$ and $N_i$ are the survivors from the previous $\Delta_i$. The initial population for the first gnomonic interval could be assumed as: (i) the number of hatching eggs (*Caddy, 1996*), (ii) the mean lifetime fecundity (*MLF*) (*Caddy, 1996*), or (iii) the number of offspring per mating event (*Lambert, 2008*).

In the gnomonic model, the estimation of $M_i$ for each $\Delta_i$ requires biological information about (i) the number of developmental stages throughout the life cycle $i \in \{1, \dots, n\}$, (ii) the duration of the first life stage corresponding to the first gnomonic interval ($\Delta_1 =$ egg stage duration), (iii) the $MLF$, and (iv) the lifespan of the species. In biological terms, the $MLF$ is inversely correlated with lifespan, except for specific groups such as semelparous populations (e.g., squids) (*Caddy, 1996*). As additional information, the duration of the other developmental stages (larvae, juvenile, adults) can be provided. The duration of the second gnomonic interval is defined as $\Delta_2 = \alpha * t_{2-1}$, where $\alpha$ is a proportionality constant, while the successive gnomonic intervals are calculated as $\Delta_i = (\alpha * t_{i-1}) + t_{i-1}$, where $i \geq 3$ up to the $i$th gnomonic interval. $M_i$ is proportional to the life stage duration since G is constant, which was expressed as follows:

$$M_i = \frac{G}{\theta_i - \theta_{i-1}}$$

where $G$ is the proportion of the overall natural death rate, being constant for all gnomonic intervals (*Caddy, 1996*), which is the product of $M_i * \Delta_i$ and $\theta_i = (\Delta_i / t_n) / 365$, representing the annual proportional duration of each interval, where $t_n$ is the longevity of the species in days (*Martínez-Aguilar, Arreguín-Sánchez & Morales-Bojórquez, 2005*).

The gnomonic approach is not applicable to marine mammals or sharks; the primary assumption of the gnomonic method is based on the estimation of natural mortality from a negative exponential function, similar to those used to estimate Z (total mortality). According to this mathematical solution, there must be abundant individuals at time 0 ($N_0$) to enable the decay in the number of individuals in the population to be estimated. Conversely, whether only one pup or individual is born, the negative exponential function cannot be solved.

## Mathematical simplification, uncertainty, and sensitivity

In this study, some equations have been modified from *Caddy (1996)* related to (i) the calculation of the duration of each subsequent gnomonic interval after the egg stage and (ii) the estimation of the constant proportion of the overall natural death rate ($G$) to improve the model performance during the computational process.

The duration of the first gnomonic interval ($\Delta_1$) is equal to the time elapsed after the moment of hatching ($t_1$). The durations of the subsequent gnomonic intervals ($i \geq 2$) are estimated as follows:

$$\Delta_i = \Delta_1 * \alpha * (\alpha + 1)^{(i-2)}$$

where

$\Delta_i$ is the duration of the gnomonic interval when $i \geq 2$;
$\Delta_1$ is the duration of first gnomonic interval $t_1$;
$\alpha$ is the proportionality constant; and
$i$ is the $i$ th gnomonic interval.

According to *Caddy (1996)* and *Martínez-Aguilar, Arreguín-Sánchez & Morales-Bojórquez (2005)*, the $\alpha$ and $G$ parameters are estimated by numerical iterations using Newton's algorithm (*Neter et al., 1996*). The $\alpha$ value is estimated by minimizing the

residuals until the sum of the durations of the gnomonic time intervals is equal to $t_n$. Unlike the iterative procedure previously used to estimate $G$, we propose an analytical solution as follows:

$$N_{i+1} = N_i * e^{-G}.$$

This equation could be expressed as follows:

$$N_n = N_0 * \left(e^{-G}\right)^n.$$

According to this equation, $G$ can be estimated based on the total number of gnomonic intervals $i \in \{1, \ldots, n\}$ and the value $N_0 = MLF$. Furthermore, $G$ is chosen so that the number of individuals surviving to the last gnomonic time interval is $N_n = 2$, following the assumption of stable population replacement with a 1:1 sex ratio; therefore, one female fulfils the requirement for population replacement if the eggs are fertilized (*Caddy, 1996*). The new equation for $G$ is expressed as follows:

$$G = -ln\left[\left(\frac{2}{MLF}\right)^{\frac{1}{n}}\right].$$

Therefore, only the constant proportionality ($\alpha$) parameter needs to be estimated. For this purpose, the *NEWUOA* numerical optimization algorithm (*Powell, 2006*; *Powell, 2008*) was used, available from the "*minqa*" package version 1.2.4 of R software (*Bates et al., 2014*; *R Core Team, 2020*).

According to *Martínez-Aguilar, Arreguín-Sánchez & Morales-Bojórquez (2005)*, the variability in $M$ was assessed assuming that $MLF$ was the main source of uncertainty, therefore simulating a total of $j$ samples of $MLF$ with a uniform distribution to determine the uncertainty of $M_i$. Then, $M_i$ estimates were obtained from $n$ simulations per gnomonic interval, obtaining the mean natural mortality rate ($\overline{M}_i$) and the standard deviation ($\sigma_{\overline{M}_i}$). Another modification in the *gnomonicM* package is the assessment of the uncertainty via a Monte Carlo simulation, which was improved via the inclusion of three assumed probabilistic density functions for $MLF$ defined as (i) uniform $MLF \sim U(MLF_{min}, MLF_{max})$, (ii) normal $MLF \sim N(\overline{\mu}_{MLF}, \sigma_{MLF})$, and (iii) triangle $MLF \sim Triangular(MLF_{min}, MLF_{max}, c_{MLF})$, where $MLF_{min}$ and $MLF_{max}$ represent the minimum and maximum of the observed $MLF$, respectively; $\overline{\mu}_{MLF}$ and $\sigma_{MLF}$ are the mean and standard deviation of the observed $MLF$, respectively; and $c_{MLF}$ is the mode of the $MLF$ in the triangular distribution. A Monte Carlo simulation must provide a correct stochastic orientation for estimating confidence intervals because the procedure (a) quantifies uncertainty based on statistical distributions derived from data rather than arbitrarily chosen distributions, (b) is unbiased, (c) is accurate, and (d) uses few distributional assumptions (*Haddon, 2011*; *Magnusson, Punt & Hilborn, 2013*). Additionally, in this study, sensitivity can be tested assuming different values in the number of gnomonic intervals, longevity, or egg stage duration in the *gnomonicM* package. The sensitivity analysis does not require assumptions regarding statistical distributions, such that the user can choose, even arbitrarily, the values of the input parameters (*Blackhart, Stanton & Shimada, 2006*; *Magnusson, Punt & Hilborn, 2013*).

## The gnomonicM package

The source code of the *gnomonicM* package is freely available from CRAN (https://cran.r-project.org/web/packages/gnomonicM/index.html) or on GitHub at https://github.com/ejosymart/gnomonicM. This package has been built to provide a user-friendly method for estimating the natural mortality ($M_i$) and temporal duration of each gnomonic interval, the $\alpha$ parameter of the gnomonic model, and the proportion of the overall natural death rate (G). The main arguments from the *gnomonicM* package are described in Table 1, and a detailed description is presented in the package manual (https://cran.r-project.org/web/packages/gnomonicM/gnomonicM.pdf). Additional print and plot methods for the deterministic and stochastic approaches are provided to show the results.

## Testing the gnomonicM package

For the application of the improved gnomonic approach and to show the functionality of the *gnomonicM* package, *gnomonicM* was tested via the deterministic method and by comparing the estimates with the results of two species reported by *Caddy (1996)*; the species had (i) seven gnomonic intervals, (ii) longevity of one year (365 days), (iii) egg stage durations of 2 days, and (iv) *MLF* values of 200,000 and 135 eggs. Additionally, the methodology was applied to published data that used the gnomonic model, and the estimates were compared with the results provided by the cited authors (see *Ramírez-Rodríguez & Arreguín-Sánchez, 2003*; *Martínez-Aguilar, Arreguín-Sánchez & Morales-Bojórquez, 2005*; *Giménez-Hurtado, Arreguín-Sánchez & Lluch-Cota, 2009*; *Martínez-Aguilar et al., 2010*; *Aranceta-Garza et al., 2016*; *Romero-Gallardo et al., 2018* for details). This approach allowed the assessment of the application of *gnomonicM* for different taxa (fish and invertebrates) and life histories (demersal, pelagic, benthic, and short and long-life spans). The *gnomonicM* package was supported by the reproducibility and verification of the results obtained from different reports, thus guaranteeing its functionality, applicability, and performance in estimating *M* for different ontogenetic developmental stages.

## The case of Pacific chub mackerel

For illustrative purposes with the entire estimation process, Pacific chub mackerel was used as an example. Thus, this section indicates the data requirements and the steps taken to apply the *gnomonicM* package, highlighting its flexibility in parameter estimations, variability in fecundity, the selected probabilistic density function, and the inclusion of auxiliary information on known gnomonic intervals different from the egg stage.

(a) *Choosing the number of gnomonic intervals*. These intervals are defined a priori and should be well represented since they are associated with the ontogenetic development stages, exhibiting realistic subunits of biological time. For Pacific chub mackerel, the biological-ecological criteria indicated the presence of eight gnomonic intervals, which are shown in Table 2. This means that the intervals should be defined following the gnomonic time framework since they are a key input in the model.

(b) *Defining the egg stage duration*. For Pacific chub mackerel, *Hunter & Kimbrell (1980)* reported that eggs of this species hatched in 56 h at 19 °C; the author also provided

**Table 1 Input arguments for the *gnomonicM* package.**

| Argument | Type | Description | Default | Required |
|---|---|---|---|---|
| nInterval | Numeric | An integer specifying the number of gnomonic intervals | – | Yes |
| eggDuration | Numeric | A single numeric value with the egg stage (first gnomonic interval) duration in days. | – | Yes |
| addInfo | Numeric vector | A numeric vector with additional information (if available) related to the observed duration of the gnomonic intervals different from the egg stage. | NULL | No |
| longevity | Numeric | An integer indicating the lifespan of the species specified in days. | – | Yes |
| fecundity | Numeric | A numeric value indicating the mean lifetime fecundity expressed as the number of eggs produced for a female. If a "normal" or "triangular" distribution is assumed, this value will be interpreted as the mean or the mode, respectively. | NULL | Yes |
| distr | Character string | Name of the probabilistic density function to be applied, which must be defined as: "uniform", "normal", "triangle". | "uniform" | Yes |
| sd_fecundity | Numeric | A numeric value indicating the standard deviation of fecundity if a "normal" distribution is assumed. | NULL | Yes |
| min_fecundity | Numeric | A numeric value indicating the minimum range of fecundity if a "uniform" or "triangle" distribution is assumed. | NULL | Yes |
| max_fecundity | Numeric | A numeric value indicating the minimum range of fecundity if a "uniform" or "triangle" distribution is assumed. | NULL | Yes |
| a_init | Numeric | A numeric value indicating the initial ($\alpha$) parameter related to the proportionality constant which will be numerically optimized. | 2 | Yes |
| niter | Numeric | An integer value representing the number of iterations. | 999 | No |
| seed | Numeric | A single value interpreted as an integer ensures that the same (pseudo) random numbers will be generated each time the script is executed. | 7388 | No |

information about the fluctuation of the elapsed hatching time as a function of temperature under controlled conditions, ranging from 33 to 117 h at 23 and 14 °C, respectively.

(c) *Including additional information if available (such as the duration of a specific cycle life stage different from the egg stage).* Occasionally, some biological information about the duration of a specific life cycle stage is reported, mainly including knowledge of early stages (e.g., the larvae stage) obtained from rearing conditions. If additional information on a known gnomonic interval (life development stage) is provided, then this time duration will not be estimated. For Pacific chub mackerel, no additional information was used in the estimation process.

(d) *Including the longevity of the species.* The concept of longevity in this study is related to the maximum age that a species could reach, and it is based on growth studies supported by reading otoliths, statoliths and other hard structures, such that the age structure of a population can be known (*Beverton, 1987*). For Pacific chub mackerel, according to *Mendo (1984)* and *Caramantin-Soriano, Vega-Pérez & Ñiquen (2008)*, a longevity of 8 years (2,920 days) was used.

**Table 2 Gnomonic intervals and the duration of each development stage based on the observed information for the Pacific chub mackerel (*Scomber japonicus*).**

| Gnomonic interval | Development stage | age (days) | | | Mean length (mm) | References[*] |
|---|---|---|---|---|---|---|
| | | start | end | elapsed | | |
| 1 | Egg | 0 | 2.3 | 2.3 | 1.05–1.14 | *Hunter & Kimbrell (1980)* |
| 2 | Pre-larvae | 2.3 | 6 | 3.7 | 2.0–3.7 | *Hunter & Kimbrell (1980)* |
| 3 | Post-larvae | 6 | 16 | 10.0 | 3.5–15.0 | *Hwang and Lee (2005)* and *Hunter & Kimbrell (1980)* |
| 4 | Early juvenile | 16 | 47 | 31 | 15–30, 24.6 | *Watanabe (1970)*; *Hwang and Lee (2005)* and *Nakayama et al. (2003)* |
| 5 | Juvenile | 47 | 150 | 103 | 30–70 | *Watanabe (1970)* and *Castro and Santana (2000)* |
| 6 | Early adult | 150 | 400 | 250 | 90–140 | *Yasuda and Hiyama (1957)* and *Castro and Lorenzo (1991)* |
| 7 | Adult | 400 | 1,063 | 663 | 140–280 | *Torrejón-Magallanes et al. (2017)* |
| 8 | Late adult | 1,063 | 2,920 | 1857 | 281–460 | *Kotlyar and Abramov (1982)*, *Castro and Santana (2000)* |

Notes.

[*]Focus on mean length (mm) and age estimates.

**Table 3 Fecundity estimations reported in the literature for the Pacific chub mackerel (*S. japonicus*).**

| Fecundity | Min | Max | SD | References |
|---|---|---|---|---|
| 78,174 | 11,805 | 144,543 | – | *Peña, Alheit & Nakama (1986)* |
| 28,978 | 7,603 | 53,921 | 1,529 | *Buitrón & Perea (1998)* |

(e) *Assigning fecundity values*. Fecundity is defined as the number of offspring per mating event (*Lambert, 2008*), and for the *gnomonicM* package, fecundity represents the population at time 0; this input can be highly variable depending on several biological, physiological, and environmental conditions (*Kjesbu et al., 1998*; *Zwolinski, Stratoudakis & Sares, 2001*; *Lambert, 2008*). Therefore, the *gnomonicM* package includes two options: the first is a deterministic approach in which the uncertainty in fecundity is ignored; the second approach involves stochasticity and includes the uncertainty of the fecundity value using a Monte Carlo simulation. For the latter option, the user must assume a probabilistic density function (uniform, normal, or triangular) linked to the fecundity. The procedure provides as outputs the precision of the natural mortality value for each gnomonic interval selected in step a). For Pacific chub mackerel, a uniform probabilistic density function was used based on the fecundity values reported by *Peña, Alheit & Nakama (1986)* and *Buitrón & Perea (1998)* (Table 3).

(f) *Assigning an initial value to the α parameter*. This step allows a statistical solution when the gnomonic model is optimized. The α parameter has a default initial value equal to 2. In cases where the information on this parameter is limited, the user should provide an acceptable α parameter according to the taxonomic group studied, using references previously reported for fishes (demersal and pelagic fishes), crustaceans (shrimp), molluscs (cephalopod and clams), or holothurians (sea cucumbers) (see Table S1). In this way, the α parameter represents an initial value that is able to prevent the algorithm from becoming stuck in local minima.

**Table 4  Results using the data from *Caddy (1996)* for two species with high fecundity MLF = 200,000 eggs and low fecundity MLF = 135 eggs.** The value in parenthesis with asterisk refers to a difference in the estimation with respect to the original work.

Longevity = 365 days
MLF = 200,000
G = 1.645
α = 1.382

| Gnomonic interval | Duration (year) | No. survivors | $M_i^-$ (year$^{-1}$) |
|---|---|---|---|
| 1 | 0.005 | 38,614 | 300.16 |
| 2 | 0.008 | 7,455 | 217.25 |
| 3 | 0.018 | 1,439 | 91.22 (*91.27) |
| 4 | 0.043 | 278 | 38.30 |
| 5 | 0.102 | 54 | 16.08 |
| 6 | 0.204 | 10 | 6.75 |
| 7 | 0.580 | 2 | 2.84 |

MLF = 135
G = 0.602
α = 1.382

| 1 | 0.005 | 74 | 109.82 |
| 2 | 0.008 | 41 | 79.48 |
| 3 | 0.018 | 22 | 33.37 |
| 4 | 0.043 | 12 | 14.01 |
| 5 | 0.102 | 7 | 5.88 |
| 6 | 0.244 | 4 | 2.47 |
| 7 | 0.580 | 2 | 1.04 |

## RESULTS

The *gnomonicM* package was tested successfully as the results obtained from previous applications were reproducible with the deterministic and stochastic approaches. The results estimated from the *gnomonicM* package using the deterministic approach and the input data provided by *Caddy (1996)* for species with high and low fecundity values are shown in Table 4 and Fig. 1. For comparative purposes between the values estimated by the *gnomonicM* package and *Caddy (1996)*, see https://cran.r-project.org/web/packages/gnomonicM/vignettes/gnomonicM.html for details.

The *gnomonicM* package with the stochastic approach was also successful when applied to other data sources, mainly for shrimp (*Farfantepenaeus duorarum*), Pacific sardine (*Sardinops caeuruleus*), and red grouper (*Epinephelus morio*); for these species, differences were not found among the parameters (G, α), durations of gnomonic intervals, or natural mortality values (Tables S1A, S1B, S1C). Conversely, for jumbo squid (*Dosidicus gigas*), white shrimp (*Litopenaeus vannamei*), and sea cucumber (*Isostichopus badionotus*), differences were found among the parameters, durations of gnomonic intervals, and natural mortality values estimated from the *gnomonicM* package. With these differences, although they varied within similar numerical scales, the results were not completely reproducible. The reasons for these difference could be explained by three sources of variability: (a) the use of additional information not provided in the reports, (b) the statistical routine

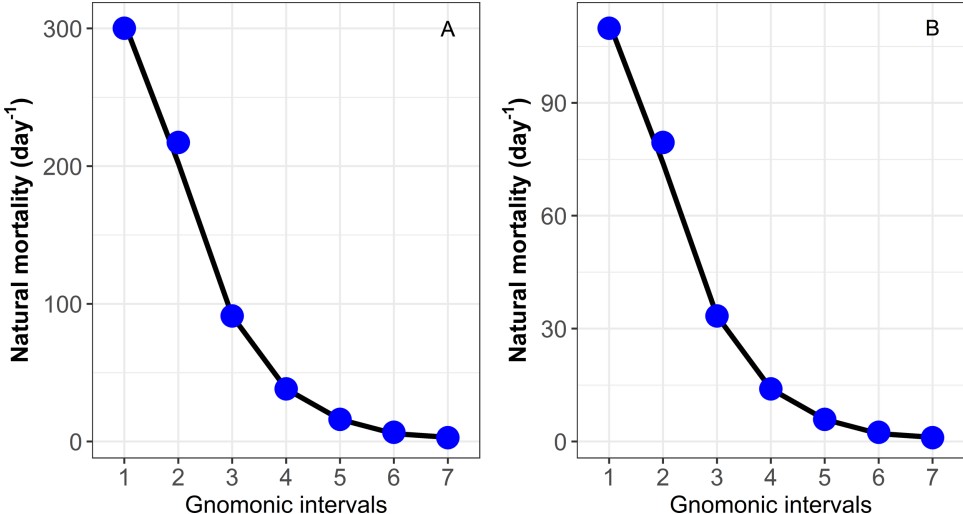

**Figure 1** **Estimation of natural mortality ($M$) by each gnomonic interval based on data provided in** *Caddy (1996).* (A) Species with higher MLF = 200,000 eggs. (B) Species with low MLF = 135 eggs. Each data point indicates the value for a particular gnomonic interval. The black line represents the rolling mean.

used for generating random numbers related to the *MLF*, or (c) possibly some parameters that could have been held constant during the numerical optimization process. The differences in natural mortality could be classified as slight for jumbo squid and white shrimp, while that for sea cucumber was greater (Tables S1D, S1E, S1F). Specifically, for *Isostichopus badionotus* the natural mortality estimates reported by *Romero-Gallardo et al. (2018)* did not include the egg stage duration as the first gnomonic interval; instead, the authors used the duration to early auricularia (planktonic larvae), and this substitution represents a misspecification in the input data, indicating methodological infringement on the gnomonic method that is easily avoidable using the *gnomonicM* package.

For the case of Pacific chub mackerel, the results based on the stochastic method are presented in Table 5. When the temporal duration of the egg stage was varied, the estimated durations of the following gnomonic intervals in comparison with their observed durations (Table 2) did not show significant differences (Kruskal–Wallis, $p = 0.94$). Comparatively, the estimates based on an egg duration of 56 h (2.33 days, $\alpha = 1.77$) were the most similar to the observed duration values. The choice of the egg stage duration influenced the estimates of $\overline{M}$, showing significant differences when comparing $\overline{M}$ values for the same gnomonic intervals (ANOVA, $p < 0.01$). The estimation of $\overline{M}_i$ and the standard deviation $\sigma_i$ decreased with age independently of the different scenarios (different egg stage durations and *MLF* values). The values of $\overline{M}_i$ for the early stages, from egg to larvae, were relatively high (344.44 yr$^{-1}$–23.87 yr$^{-1}$), while the $\overline{M}_i$ values for adults ($\overline{M}_7 = 0.62$ yr$^{-1}$ –0.73 yr$^{-1}$) and late adults ($\overline{M}_8 = 0.22$ yr$^{-1}$ –0.27 yr$^{-1}$) showed low variability and the lowest values (Fig. 2, Table 5). The parameter $\alpha$ was associated with the durations of life-history stages, showing an inverse relation with the egg stage duration, and it was independent of the *MLF*;
the *G* parameter had a constant value under three different egg stage duration scenarios, and its magnitude increased with greater *MLF* values (Table 5).

## DISCUSSION

### The gnomonicM package and other methods

Compared to the method employed by *Caddy (1996)*, the *gnomonicM* package uses simplified algebra, increasing its parsimony. The original gnomonic model proposed estimations of the *α* and G parameters in two independent numerical optimizations, while the *gnomonicM* package used an algebraic estimator for G and an analytic estimator for *α* supported by the *NEWUOA* numerical optimization algorithm (*Powell, 2006*; *Powell, 2008*). Another feature of the *gnomonicM* package is the biological and ecological sense linking the gnomonic concept to the life cycle of any given species, expressed as the development of the ontogenetic stages (*Martínez-Aguilar, Arreguín-Sánchez & Morales-Bojórquez, 2005*). Moreover, the *gnomonicM* package allows the incorporation of auxiliary information for specific gnomonic intervals related to their observed time durations; this contribution is useful for calibrating estimates of *M* vectors. Furthermore, the *gnomonicM* package provides confidence intervals (CI 95%) of natural mortality for each estimated gnomonic interval, assuming that the main source of uncertainty is the mean lifetime fecundity. The latter can be assumed to be non-informative (i.e., with a uniform distribution) or informative (i.e., with normal and triangular distributions). The choice of the probabilistic density function will affect the natural mortality estimates; however, the uncertainty regarding fecundity must not be limited to only non-informative distribution (*Martínez-Aguilar, Arreguín-Sánchez & Morales-Bojórquez, 2005*).

According to *Quinn & Deriso (1999)* and *Kenchington (2014)*, the natural mortality estimation methods require different input data and variables, and some of these data could be difficult to obtain because they are usually associated with extensive time series (i.e., length-frequency analysis and some stock assessment models); others, such as tag-recapture and telemetry in oceans, are usually expensive and sometimes provide imprecise estimates (*Hearn, Pollock & Brooks, 1998*; *Pine et al., 2003*; *Pollock, Jiang & Hightower, 2004*), while some are supported by knowledge of the age structure of a population and its growth parameters (in methods based on the life history and maximum observed age of a species). These meta-analysis estimators are biological generalizations that use multiple regression analysis with independent variables, such as growth and environmental variables, whose application is limited to certain marine taxa. Regarding the uncertainty, a common feature of the above methods is the absence of an uncertainty calculation, expressed as a confidence interval (*Kenchington, 2014*). Although the *gnomonicM* package requires specific biological data, it provides estimates of *M* for the entire life cycles of marine organisms; estimations are commonly scarce for egg and larval stages, and the *gnomonicM* package also includes an analysis of uncertainty represented by the confidence intervals of *M*. Furthermore, the sensitivity of the estimated *M* values can be tested by varying the duration of the egg stage, since this stage is a critical input data for the gnomonic model, affecting the estimated *M* values mainly at the early stages of development. Finally, the *gnomonicM* package provides

Torrejón-Magallanes et al. (2021), *PeerJ*, DOI 10.7717/peerj.11229

**Table 5   Estimates of natural mortality ($M$) and durations for each gnomonic interval for the Pacific chub mackerel (*Scomber japonicus*) with different MLFs (assuming a uniform distribution), and egg stage durations.**

Longevity = 2,920 days
MLF = 78,174 [11,805–144,543]
$G = 1.30$

| Stage of development | $\alpha = 1.99$ | | | $\alpha = 1.77$ | | | $\alpha = 1.49$ | | |
|---|---|---|---|---|---|---|---|---|---|
| | Duration (days) | $M_i$ (year$^{-1}$) | $\sigma i$ | Duration (days) | $M_i$ (year$^{-1}$) | $\sigma i$ | Duration (days) | $M_i$ (year$^{-1}$) | $\sigma i$ |
| Egg | **1.38** | 344.44 | 0.05750 | **2.33** | 202.98 | 0.03389 | **4.88** | 97.15 | 0.01622 |
| Prelarvae | 2.73 | 173.31 | 0.02893 | 4.13 | 114.67 | 0.01914 | 7.28 | 65.06 | 0.01086 |
| Postlarvae | 8.16 | 58.01 | 0.00968 | 11.44 | 41.40 | 0.00691 | 18.15 | 26.09 | 0.00436 |
| Early juvenile | 24.39 | 19.42 | 0.00324 | 31.69 | 14.94 | 0.00249 | 45.25 | 10.46 | 0.00175 |
| Juvenile | 72.86 | 6.50 | 0.00109 | 87.79 | 5.40 | 0.00090 | 112.83 | 4.20 | 0.00070 |
| Early adult | 217.66 | 2.18 | 0.00036 | 243.17 | 1.95 | 0.00033 | 281.33 | 1.68 | 0.00028 |
| Adult | 650.25 | 0.73 | 0.00012 | 673.59 | 0.70 | 0.00012 | 701.43 | 0.68 | 0.00011 |
| Late adult | 1,942.57 | 0.27 | 0.00004 | 1,865.86 | 0.25 | 0.00004 | 1,748.86 | 0.24 | 0.00005 |

MLF = 28,978 [7,603–53,921]

$G = 1.19$

| Stage of development | $\alpha = 1.99$ | | | $\alpha = 1.77$ | | | $\alpha = 1.49$ | | |
|---|---|---|---|---|---|---|---|---|---|
| Egg | **1.38** | 315.16 | 0.04733 | **2.33** | 185.72 | 0.02789 | **4.88** | 88.89 | 0.01335 |
| Prelarvae | 2.73 | 158.58 | 0.02381 | 4.13 | 104.92 | 0.01576 | 7.28 | 59.53 | 0.00894 |
| Postlarvae | 8.16 | 53.08 | 0.00797 | 11.44 | 37.88 | 0.00569 | 18.15 | 23.87 | 0.00359 |
| Early juvenile | 24.39 | 17.77 | 0.00267 | 31.69 | 13.67 | 0.00205 | 45.25 | 9.57 | 0.00144 |
| Juvenile | 72.86 | 5.95 | 0.00089 | 87.79 | 4.93 | 0.00074 | 112.83 | 3.84 | 0.00058 |
| Early adult | 217.66 | 1.99 | 0.00030 | 243.17 | 1.78 | 0.00027 | 281.33 | 1.54 | 0.00023 |
| Adult | 650.25 | 0.67 | 0.00010 | 673.59 | 0.64 | 0.00010 | 701.43 | 0.62 | 0.00009 |
| Late adult | 1,942.57 | 0.25 | 0.00003 | 1,865.86 | 0.23 | 0.00003 | 1,748.86 | 0.22 | 0.00004 |

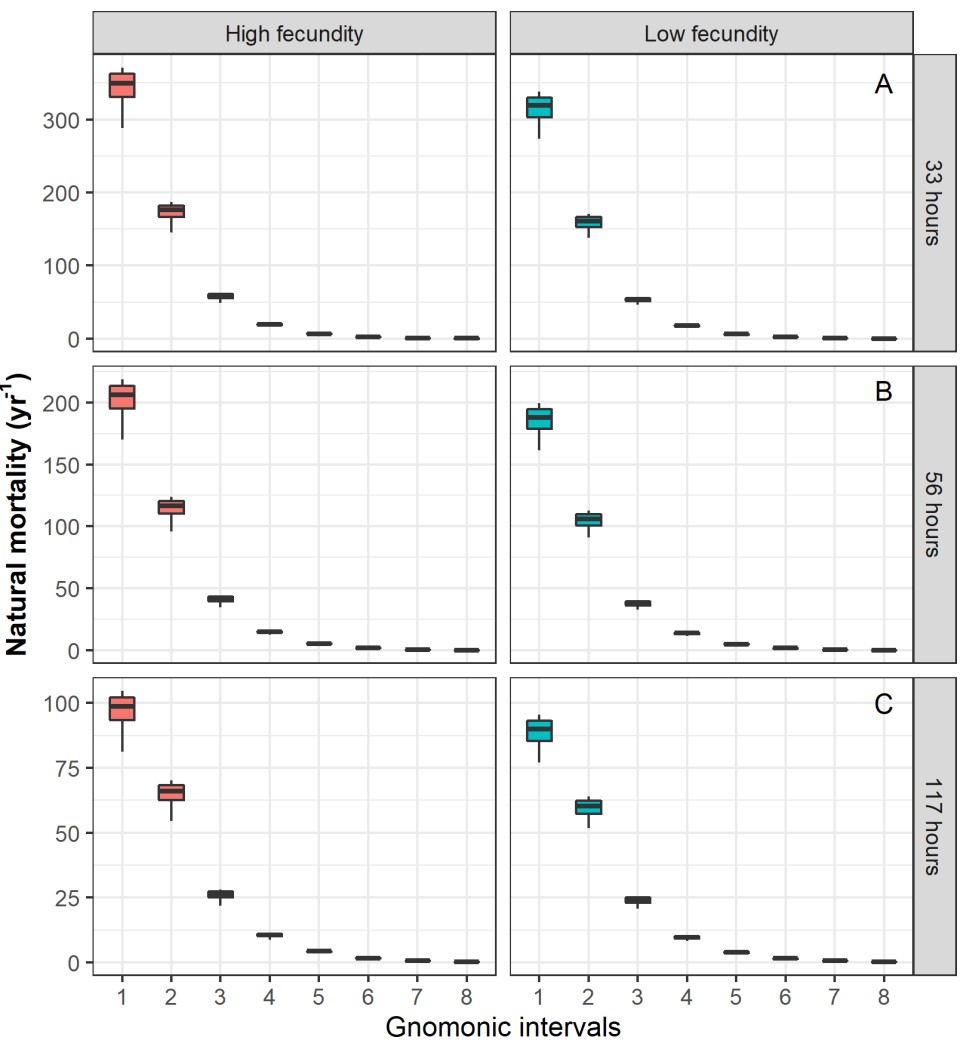

**Figure 2** Natural mortality ($M$) vector estimated for Pacific chub mackerel (*Scomber japonicus*). Columns represent a species with higher MLF = 78,174 and lower MLF = 28,978 values. Rows represent a different egg stage duration: (A) 33 h, (B) 56 h, (C) 117 h, respectively.

estimations of $M$ via numerical optimization in comparison to deterministic estimators (see *Kenchington, 2014*).

## The Pacific chub mackerel case and details about gnomonicM applications

The aim of presenting a new case study was to provide a comprehensive guide for data collection and obtaining results and to explain the details of the application of the *gnomonicM* package to avoid its misuse. Based on the gnomonic approach, biological information regarding the life cycle of a given species must include the temporal duration of the egg stage, longevity, and fecundity. For Pacific chub mackerel, the biological data were collected from several sources (Table 1). At this step, the main challenge was the different criteria among authors focusing on the definitions of biological stages and their

durations. For example, in the larval stage, it is common to find subclassifications, referred to as pre-larvae, early larvae, larvae, yolk-sac larvae, and post-larvae stages; for some species, there are even subcategories defined as I or II (e.g., *Martínez-Aguilar, Arreguín-Sánchez & Morales-Bojórquez, 2005*; *González-Peláez et al., 2015*). Moreover, contrasting results regarding the development stage can be documented from rearing conditions or field observations. The solution to this problem can be addressed by the well-defined concept of the gnomonic interval (*Caddy, 1991*; *Caddy, 1996*). This means that some stages could be considered individually or grouped to make up a particular gnomonic interval considering the biological and ecological characteristics of the interval, with the purpose that each successive gnomonic interval is greater than the previous interval in its duration. If this condition is not satisfied, then the results will be biased due to the misspecification of data. Another cause can be related to the lack of specific biological data for the species studied; therefore, some generalizations could be assumed, such as, for example, using a characteristic (e.g., duration) of the development stages of other members of the same taxonomic genus, even from different geographical regions.

Once the data have been collected, they are introduced directly as input arguments (see Table 1). These inputs are simply composed of numbers, vectors, or characters depending on the approach used, either deterministic or stochastic, providing a quick, flexible, and user-friendly tool with characteristics that a software package should have (*Wilson et al., 2017*). This approach allows users to focus on obtaining and interpreting results rather than the calculation process. Therefore, when the deterministic approach is selected via the gnomonic function, the input data must be organized as follows:

```
x <- gnomonic(nInterval = 8,
              eggDuration = 2.33,
              longevity = 2920,
              fecundity = 78174,
              a_init = 2)
```

In the case of the stochastic approach via the gnomonicStochastic function, the input data must be organized as follows:

```
x <- gnomonicStochastic(nInterval = 8,
              eggDuration = 2.33,
              longevity = 2920,
              min_fecundity = 11804,
              max_fecundity = 144543,
              distr = "uniform",
              niter = 1000,
              a_init = 2).
```

For any option selected previously, the numerical outputs can be obtained from print(x), and the graphic representation is available for the user from plot(x).

The scenarios described above represent a sensitivity analysis available from the *gnomonicM* package; the rationale is supported by the role of the environmental variability influencing the fecundity and the egg stage duration, both of which have impacts on the natural mortality of *S. japonicus*. The sensitivity analysis must be based on biological and

**Table 6 Estimates and assumed values of natural mortality ($M$) for the Pacific chub mackerel (*Scomber japonicus*) based on different methods.**

| Stage of development | M (year$^{-1}$) | Area | Method | Reference |
|---|---|---|---|---|
| Fish larvae | 51.1 | Japan | Rearing conditions | *Watanabe (1970)* |
| Adults | 0.52–0.53 | Peru | Pauly estimator | *Caramantin-Soriano, Vega-Pérez & Ñiquen (2008)* |
| Adults | 0.5 | California current | Regression of Z on F | *Parrish and MacCall (1978)* |
| Adults | 0.5 | Eastern Central Pacific | *Regression of Z on F | *Patterson et al. (1993)* |
| Adults | 0.5 | Southern California - Northern Baja California | *Regression of Z on F | *MacCall et al. (1985)* |
| Juveniles - Adults | 1.01 | Gulf of California | Length frequency data | *Cisneros et al. (1990)* |
| Adults | 0.5 | Japan | *Empirical equation | *Yatsu et al. (2002)* |
| Juvenile - Adults | 0.81 | U.S.A, Mexico | Statistical catch at age | *Crone et al. (2019)* |

Notes.
*Assumed value.

ecological knowledge of the analysed species, such that the scenarios estimated from this approach can be plausible. In this way, the use of the *gnomonicM* package requires clear criteria for estimating the number of scenarios useful for each case. In this study, we tested scenarios based on the duration of the egg stage and *MLF*. We chose the scenario of 56 h (2.33 days) as the "best" scenario because the estimations of the durations of the gnomonic intervals obtained under this scenario were very similar to the observed values (Tables 2 and 5). Additionally, these durations increased over the life cycle following the gnomonic time concept (*Caddy, 1996*). The *M* values for different stages were similar to the reported values for the egg and larvae stages (*Watanabe, 1970*; *Ware & Lambert, 1985*), and the differences were presented for the adult phases due to the method used, such as catch curve analysis, stock assessment and empirical equations (Tables 5 and 6).

The natural mortality values estimated from the *gnomonicM* package for Pacific chub mackerel showed an adequate biological trajectory through the different ontogenetic developmental stages, with the highest values observed during the early development stages, indicating that the gnomonic times selected for this species were adequately grouped. Regarding ecological theory, the early stages of Pacific chub mackerel are exposed to high rates of predation by planktonic organisms and fish. This species has a variety of predators depending on the phase of development. The main predators at different sizes of Pacific chub mackerel, including juveniles, are hake *Merluccius gayi* and the eastern bonito *Sarda chilensis* (*Ojeda & Jaksic, 1979*; *Fuentes, Antonietti & Muck, 1989*). Cannibalism is another source that increases natural mortality, and several studies on this species have shown evidence of cannibalism in eggs and individuals 8-mm and larger, particularly in spawning grounds. It is important to mention that if the egg stage duration is long or if the transition from egg to larvae is slow, the organism may be subject to predation mortality for longer periods, resulting in high natural mortality (*Pitcher & Hart, 1982*).

Finally, the improved approach and the use of the *gnomonicM* package adequately represented the biological developmental stages of the species assessed in this study over its life cycle. Additionally, estimates of *M* produced by the *gnomonicM* package may provide
basic input data to ecological models, enabling them to use estimates of variable $M$ across different ages groups, mainly for the life stages affected by fishing.

## CONCLUSIONS

The improved gnomonic approach implemented in the *gnomomicM* R package provides a method for estimating natural mortality ($M$) throughout the different life stages of aquatic species, fish, and invertebrates. The data required for *gnomonicM* must include, at least, the number of gnomonic intervals, egg stage duration, longevity, and fecundity. These input data have intrinsic uncertainty: (a) fecundity is associated with high levels of uncertainty because the mean lifetime fecundity depends on specific studies on reproductive biology and is influenced by environmental variability, density-dependent effects, and biological features of the stock; (b) the egg stage duration is commonly taken from rearing studies under controlled conditions (e.g., temperature, salinity) as the duration can rarely be obtained from field data; (c) the longevity depends on the maximum age identified from the age structure for the population studied; thus, the longevity requires specific studies of hard structures (e.g., otoliths, spines, vertebrae); the number of gnomonic intervals must be determined, these must be established using biological criteria when using the gnomonicM package, such that the ontogenetic development stages can be useful in reducing the uncertainty in the number of intervals selected. The number of gnomonic intervals provided must be such that the duration of each successive gnomonic interval is greater than the previous one ($\Delta_i$), which implies that several development stages can be grouped to satisfy this assumption. Specifically, the last gnomonic interval would group the ontogenetic stage of adult individuals; this criterion enables this interval to be highly flexible. For annual-lived species several months could constitute a group (the lifetime for adults), while for long-lived species several years could be grouped, these being linked to the age structure of the adult population. Additionally, *gnomonicM* allows the incorporation of auxiliary information related to the observed temporal duration of specific gnomonic intervals, which is useful for calibrating estimates of $M$ vectors. Finally, the additional `plot()` and `print()` functions are provided for numerical and graphical representations, making the package quick, flexible, and easy to use and allowing users to focus on obtaining and interpreting results rather than on the calculation process.

## ACKNOWLEDGEMENTS

Josymar Torrejón-Magallanes wishes to thank Miguel Ñiquen and Marilú Bouchon for the biological information and discussion about the life history of Pacific chub mackerel and Enrique Ramos for mathematical support. We sincerely wish to thank the editor and two anonymous reviewers for their thoughtful comments and suggestions that improved both the quality of this manuscript and the *gnomonicM* package.

### Funding

Josymar Torrejón-Magallanes was supported by Consejo Nacional de Ciencia y Tecnología México (CONACyT) for the PhD fellowship CVU 388649. Francisco Arreguín-Sanchez was supported by Instituto Politécnico Nacional through SIP-20201413 project, the EDI and COFFA programmes. There was no additional external funding received for this study. The funders had no role in study design, data collection and analysis, decision to publish, or preparation of the manuscript.

### Grant Disclosures

The following grant information was disclosed by the authors:
Consejo Nacional de Ciencia y Tecnología México (CONACyT) for the PhD fellowship CVU 388649.
Instituto Politécnico Nacional through SIP-20201413 project.
The EDI and COFFA programmes.

### Competing Interests

The authors declare there are no competing interests.

### Author Contributions

- Josymar Torrejón-Magallanes conceived and designed the experiments, performed the experiments, analyzed the data, prepared figures and/or tables, authored or reviewed drafts of the paper, build and upload the package to the R CRAN and GitHub, and approved the final draft.
- Enrique Morales-Bojórquez and Francisco Arreguín-Sánchez conceived and designed the experiments, analyzed the data, authored or reviewed drafts of the paper, and approved the final draft.

### Data Availability

Data is available at:
gnomonicM: Estimate Natural Mortality for different life stages: https://cran.r-project.org/web/packages/gnomonicM/index.html.
GitHub, gnomonicM: Available at https://github.com/ejosymart/gnomonicM
Vignette: Available at https://cran.r-project.org/web/packages/gnomonicM/vignettes/gnomonicM.html

### Supplemental Information

Supplemental information for this article can be found online at http://dx.doi.org/10.7717/peerj.11229#supplemental-information.

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
