# Peer review of "Improving the gnomonic approach with the gnomonicM R-package to estimate natural mortality throughout different life stages"

_PeerJ, doi:10.7717/peerj.11229_

## Round 0.1 · original submission · Major Revisions

Dear Authors,

As you can see two reviewers provided detailed comments on your manuscript. One of them suggested its rejection, while the second one considered the manuscript potentially suitable after revisions. In particular, the latter reviewer suggested that the paper should undergo some minor changes, as it should not take too much effort to respond to its concerns, even if some of them are quite substantial.

I will reconsider the manuscript after the revision, and I would suggest you focus on the points raised by both reviewers related to the assumptions underlying the proposed method, and the comparison with other approaches, more widely used in fishery sciences. I would also ask you to carefully prepare a rebuttal letter, explicitly addressing also the comments on which you disagree or for which it is not possible to positively answer.

Reviewer 1 ·

Basic reporting

see general comments

Experimental design

see general comments

Validity of the findings

see general comments

Additional comments

In this paper the authors present a package for R to estimate stage-dependent natural mortality (M) using the gnomonic approach developed by John Caddy. They present a simplified approach for estimation and compare the results obtained using the package to those obtained previously using other optimization methods. They also compare their results with stage dependent duration and M values for Pacific chub mackerel.

In general I found the paper to be reasonably well written, although it will require some native English language editing which I do not provide here. The description of the gnomonic approach is not entirely clear and I had to refer to the original articles by Caddy to understand the approach. This approach does not have wide use in fisheries science. While I realize that the present article is largely an application of previously published research, I question some of the validity of that research in my comments that follow as they affect the validity of the proposed R package.

First, while there is a clear benefit to modelling M as age or stage dependent, I question the benefits of trying to model both early life and adult mortality in one approach. Mortality during early life is known to be highly variable particularly for fecund marine taxa, on time scales that are much shorter (annual) than variation in MLF (e.g., Houde 1987). At best the gnomonic approach can provide an approximation of that mortality. That high variation is why the vast majority of stock assessment and ecosystem models model recruitment to a more advanced age (juvenile or later) where mortality is lower and less variable.

Second, the approach is presented as somehow different and perhaps advantageous over other estimators of M which are characterized by the authors as being variable and potentially biased. However, ultimately it uses the same information as many of those approaches, namely longevity. In that sense it differs only a little from estimators proposed based on life-history theory (Charnov) or empirical relationships based directly on longevity (e.g., Hoenig) or indirectly via asymptotic length and growth rate (e.g, Pauly, Ghislason et al., Then et al.).

Third, the authors assume, based on assumptions in a previous paper, that variability in stage-dependent M is driven entirely by variation in fecundity. This is simply not demonstrated. It is well known in demography that changes in mortality will affect longevity unless there is a compensatory response in survival among life stages. Thus to estimate variability in mortality using an estimator that is based in large part on longevity, one needs to consider variability in that parameter. Furthermore, it is also well known that egg and larval stage duration can be quite variable, particularly for many invertebrates with long pelagic stages. Based on the eqn on line 140 is seems that variation in delta1 is also very pertinent.

Fourth, notwithstanding the previous point, nothing shown previously or in this paper demonstrates that the gnomonic approach is a proper estimator of M. Yes, it appears to be able to capture stage/age dependent patterns to some extent, but there has been no extensive evaluation to demonstrate this across multiple taxa and to characterize the bias and variation involved. Caddy’s original work is largely based on theoretical examples, noting that the present author’s ability to reproduce those results only validates the calculation not the validity of the method for estimating M for wild populations. Certainly the data exist for more than just chub mackerel to undertake a proper evaluation of the method. However, even for that species it is clear from the results reported by the authors that the approach does not capture adult, sub-adult and juvenile mortality well (based on the tables biases range from 35-400%). Notably these are the stages for which knowledge and assumptions on M is absolutely critical for stock assessments; early life and juvenile mortality is generally subsumed into ‘recruitment variability’ in these models.


Fifth, the Monte Carlo simulations used by the authors to simulate variability in M are neither theoretically or empirically demonstrated as providing a proper simulation of sampling variability. It is therefore highly misleading to claim that the R package can produce proper confidence intervals on M; ie. that 19 times out of 20 the interval will contain the true M value. Simulated variability is fine to show sensitivity to input assumptions but creating confidence intervals creates an inherent expectation of credibility that is simply not supported.



Other comments:

Abstract – The gnomonic approach is not sufficiently well known to assume that the reader of the abstract will know what it is, nor does the name provide an intuitive idea of what it may represent. The authors should briefly define it.
-Reader will not know what the proportionality constant and the proportion of overall natural death rate are.
-The acronym MLF is not defined.
-The sentence beginning on line 31 is complex and difficult to comprehend. Amonst other things, the reference to a constant M isn’t clear and could likely be interpreted as constant in time rather than common across life stages.


L69 – This is a bit of a mischaracterization. Caddy proposes a method for estimating the stage-dependency of M, really as a discretization of a continuous size-dependent process. He does not claim to be estimating M itself.

L140 – I am not certain that I understand the equation, particularly why there is alpha plus one in the brackets. It seems to me that the right side of the equation should be delta1*alpha^(i-1)

L204-206 – This point is much overstated. As described in my comments above, there is no guarantee of applicability and performance. Also I do not see how this package can ‘provide a good understanding of the biological importance of M’; nothing presented here shows this regardless of the method used to estimate M.

L266+ It is not clear how the reader is meant to interpret the supplement.

L276 – what does ‘mainting some fixed parameters’ mean?

L305 – The statement ‘usually when statistical models are simplified, there is an improvement in their performance’ is unsupported and strictly speaking not true. Parsimony is important, not mere simplicity.

Lines 408-410 – This is not an emergent result, the method is specifically designed to produce the highest values during early stages


Reference

Houde, E. 1987. Fish early life dynamics and recruitment variability. Am Fish Soc Symp 2: 1729

Reviewer 2 ·

Basic reporting

No comment

Experimental design

No comment

Validity of the findings

No comment.

Additional comments

This paper simplifies the use of the gnomonic approach developed in a couple of papers by Caddy to estimate natural mortality (M). I must confess that I was not familiar with this approach to estimate M and therefore cannot comment too profoundly on its theoretical basis and its implicit and explicit assumptions. Nevertheless, I find the approach interesting and the current paper of value, especially for estimating M in short-lived marine organisms. The development of the R package to apply the method certainly adds to the value of the paper. I provide detailed comments below that the authors should address prior to final acceptance of the document. Briefly stated here, my main issues relate to: 1) some of the method’s assumptions; 2) the calculation of G and other computational issues; and 3) the inability of the reader to judge the model validation owing to the lack of simultaneous presentation of results from the current work and published papers.


Detailed comments (line numbers correspond to those in the Word version of the ms):

Lines 112-114: how reasonable is the assumption that for the first gnomonic interval 〖(∆〗_1) the number of hatching eggs (initial population) is equivalent to the mean lifetime fecundity (MLF)?
Lines 117-118: What about for life-bearing fish, such as sharks for example that give birth to fully formed pups? What would you consider the initial stage equivalent to “eggs” to be? Age-0s?
Lines 118-119: MLF is inversely correlated with lifespan.
Line 124: So the M is proportional to the life stage duration since the G is constant.
Line 157: Shouldn’t the “n” superscript be “i” since the number of survivors will vary by life stage?
Lines 160-162: what is the rationale for using 2? 1 male and 1 female?
Line 163: so the G (constant proportion of M) is a function of the number of life stages and lifetime fecundity only. Unless I’m missing something I get a different value with the first example in the vignette (vignette=1.644; I get 14.459 with MLF=200000 and n=7)?
Line 170: the n here refers to the number of iterations but it is also used for the number of gnomonic intervals, which is confusing.
Lines 182-184: Note that the example in the R vignette for high and low fecundity has the same fecundity value. Change the LFM to 135 from 200000 for “model_lf”.
Lines 231-232: in the very first example in the vignette where the duration of the second and fifth gnomonic intervals was set equal to 4 and 40 days respectively, the output of the model still estimated stage duration (at 2.8 and 37 days). Why?
Lines 235-237: one can argue that longevity is more directly related to M than fecundity and it is often poorly known. It would be advisable to incorporate uncertainty in lifespan in the stochastic portion of the package.
Lines 253:256: in the first example given in the vignette, the initial value given to the estimated proportionality constant (alpha) does not seem to affect results at all (I changed it from 2 to 3 and to 0.2 and results did not vary at all). So how important is it to correctly specify an initial value of this parameter, which is the only one estimated in the optimization? Also, how would one go about setting an initial value for this proportionality constant if none exists?
Lines 264-266: I don’t see a comparison of the results obtained with this package and those from Caddy in the online vignettes referenced here. According to Table 4 the results were identical except for the M stage 3 value varying in the second decimal.
Lines 267-273: But Table S1 only shows results obtained with the package so we have to take the authors’ statements at face value. All the tables shown should include a comparison of values from the package and the original studies.
Lines 286-290: I don’t understand what’s being conveyed here. I only see estimated time durations of the stages, not observed ones.
Line 326: Most stock assessment models fix M because there is not enough information in the data to estimate it, even for data-rich stocks.
Lines 347-349: In your examples stage duration is most often estimated. So be explicit in saying that egg state duration is a requirement, but the others are not.
Lines 357-361: if increasing stage durations is a pre-requisite of the gnomonic approach then its application could be impaired in some cases, like for example if it were to be applied to long-lived vertebrates such as sea turtles or sharks that may have a subadult stage shorter than the large juvenile stage. Discuss this potential limitation of the method.
Lines 401-402: should be Tables 2 and 5.
Lines 409-411: but this is dictated by the method itself since M is inversely proportional to the stage duration and the nominator in the equation in line 124 (G) is a constant.
Line 424: again, it’s not possible to check if these statements are true because a comparison between results of the package and those from the original studies was not provided.
Lines 429-431: Include the caveat here about long-lived species and the uncertainty in estimates of longevity. Of the species included for “model validation” the longest living one is the grouper with 20 years but the majority are short-lived invertebrates. A caveat should be given about the applicability of the method to long-lived species, especially given that the method assumes the main source of uncertainty in estimating M comes from fecundity, whereas lifespan is very important for many fish species for example, and is also highly uncertain.


Minor comments:

Line 23: define MLF in abstract.
Line 46: add “size” as well.
Lines 55-59: Most of these methods estimate Z, from which F is estimated by subtracting an assumed M from Z.
Line 245: bootstrapping or Monte Carlo simulation?
Lines 256-257: I would express “to avoid the stagnation of the statistical optimizer” a bit differently. Something like “to prevent the algorithm to get stuck in local minima”.
Line 331: independent variables.
Lines 337 and 394: it’s “sensitivity”, not “sensibility”.
Lines 337-340: rewrite as the meaning of this sentence is not clear.
Lines 406-407: most methods shown in Table 6 for adults are based on a catch curve, not on empirical estimators.

---

## Round 0.2 · accepted · Accept

I think that the extensive responses to the comments and the integration and clarification into the text make the manuscript ready for publication, as it easier for the reader to appreciate the strength and limitations of the proposed approach.

Reviewer 2 ·

Basic reporting

N/A

Experimental design

N/A

Validity of the findings

N/A

Additional comments

N/A